# IN-CONTEXT META LEARNING INDUCES MULTI-PHASE CIRCUIT EMERGENCE

**Gouki Minegishi**[1]  **Hiroki Furuta**[1]  **Shohei Taniguchi**[1]  **Yusuke Iwasawa**[1]  **Yutaka Matsuo**[1]
[1]The University of Tokyo
{minegishi,furuta,taniguchi,iwasawa,matsuo}@weblab.t.u-tokyo.ac.jp

## ABSTRACT

Transformer-based language models exhibit In-Context Learning (ICL), where predictions are made adaptively based on context. While prior work links induction heads to ICL through phase transitions, this can only account for ICL when the answer is included within the context. However, an important property of practical ICL in large language models is the ability to meta-learn how to solve tasks from context, rather than just copying answers from context; how such an ability is obtained during training is largely unexplored. In this paper, we experimentally clarify how such meta-learning ability is acquired by analyzing the dynamics of the model's circuit during training. Specifically, we extend the copy task from previous research into an In-Context Meta Learning setting, where models must infer a task from examples to answer queries. Interestingly, in this setting, we find that there are multiple phases in the process of acquiring such abilities, and that a unique circuit emerges in each phase, contrasting with the single-phase transition in induction heads. The emergence of such circuits can be related to several phenomena known in large language models, and our analysis lead to a deeper understanding of the source of the transformer's ICL ability.

## 1 INTRODUCTION

Transformer-based language models (Vaswani et al., 2017) show an intriguing ability to perform In-Context Learning (ICL) (Brown et al., 2020; Xie et al., 2021; Garg et al., 2022; Dong et al., 2024). ICL is the ability to predict the response to a query based on context without any additional weight updates. A widely adopted application of ICL is *few-shot learning* in which only a small number of examples in the context guide the model's response to a new query. Due to its unique capability, ICL has gained a lot of attention in the research community, and there have been several approaches such as Bayesian inference (Xie et al., 2021) and meta-gradient descent (Von Oswald et al., 2023) to uncover its underlying mechanisms.

One of the popular approaches to understanding ICL is mechanistic interpretability: reverse-engineering the computations performed by models (Elhage et al., 2021). A key focus within this framework is the study of *circuits*, subgraphs with distinct functionality that serve as fundamental building blocks of neural network behavior (Wang et al., 2022; Conmy et al., 2023a). Notably, Olsson et al. (2022) uncovered *induction heads*, a specific circuit mechanism that plays a crucial role in enabling ICL. Induction heads recognize the repeating pattern [A][B] ... [A] within the context and predict [B] as the next token through a match-and-copy operation (Figure 1-(a)). The existence of induction heads is further investigated under more complex tasks, such as performing semantic matching (Ren et al., 2024), serving as subcomponents of circuits for natural language tasks within LLMs (Wang et al., 2022; Merullo et al., 2024), and engaging in intricate interactions with multi-head attention (Singh et al., 2024).

However, the copy mechanism as described in the induction head explains only a fraction of the few-shot ICL. Let us consider, for instance, the following ICL scenario in a Country-to-Capital task, based on Hendel et al. (2023):

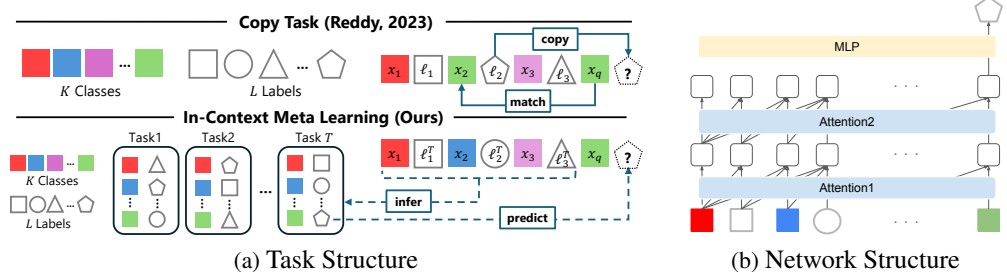

(a) Task Structure           (b) Network Structure

Figure 1: (a) **Task Structure**: Previous studies focused on a copying-task setup, where the query's answer remains unchanged by context, allowing the model to either memorize pairs or *match* and *copy* from context. In contrast, this work explores a more practical scenario where $(x, \ell)$ pairs vary by task, requiring the model to *infer* the task from examples and *predict* the query's answer. (b) **Network Structure**: we mainly use two layers of attention followed by a token-wise MLP layer. The task is consistent within the context.

$$\underbrace{\text{France} \to \text{Paris}, \quad \text{Spain} \to \text{Madrid},}_{\text{example}} \quad \underbrace{\text{Japan} \to}_{\text{query}} \quad \underbrace{?}_{\text{prediction}}$$

It is well known that ICL can enhance performance in this scenario however, this improvement cannot be explained merely by retrieving similar examples through induction heads. A straightforward way to explain this ability is to assume that the model infers the task from the examples and then uses this inferred task to make predictions. For example, Hendel et al. (2023); Todd et al. (2024) demonstrates that tasks are internally represented as vectors (i.e., task vectors) within the LLM. This task inference ability is recognized as a form of meta-learning (Min et al., 2022a). However, it remains unclear exactly what kind of circuit implements this meta-learning capability or how the circuit is acquired.

In this study, our goal is to elucidate how such meta-learning capability is acquired. To that end, we extend the copy task from previous research (Reddy, 2023) to a problem setting, which we call In-Context Meta-Learning (ICML) setting, that requires task inference. We then train a simplified transformer on this extended setting, and analyze changes in its internal circuits during the training process. In this setting, as shown in Figure 1-(a), there exists a set of multiple tasks, and the answers differ from task to task, so the model needs to infer the task from the examples to answer the query. Interestingly, we observe that learning dynamics emerge in this setting that differ significantly from the case of simple copying tasks. First, as shown in Figure 2, we find that the model undergoes three phase transitions while acquiring meta-learning capabilities, unlike the single transition typically observed in copying tasks. More specifically, we find that in the first phase, a bigram-type circuit emerges that focuses solely on the query, ignoring the context and relying only on the model's weights. In the second phase, a circuit emerges that pays attention only to the labels in the context. Finally, a circuit emerges that chunks each example pair into a single token.

We introduce novel metrics to systematically measure these three circuits and show that the abrupt change of these metrics aligns closely with the sudden jumps in accuracy. Notably, the label-focused circuit that emerges in the second phase suggests that during acquiring meta-learning capabilities, the model may initially learn to identify tasks by examining only the set of labels, without considering the correspondence between classes and labels. The existence of the label-focused circuit also corresponds to the phenomenon in previous studies (Min et al., 2022b) that LLMs maintain high ICL performance even under random label assignments, which is one explanation for the unique nature of LLMs.

We also examine the case of a multi-head model, which is a more practical setting; phase transitions in accuracy become less apparent, and different heads can still specialize in *parallel* — for instance, one head may converge on a particular circuit, while another becomes a different one. Although this parallel specialization leads to smoother accuracy improvements, our circuit-level metrics uncover hidden circuit emergence, revealing that even though phase transitions remain invisible in the accuracy curve, the underlying circuits still change abruptly. This observation suggests that even when a clear phase transition is not observed on the loss curve, as in the case of LLM training, phase transitions

can occur on the circuits, which leads to bridging the gap between toy experiments in the study of mechanistic interpretability and practical scenarios.

## 2 RELATED WORKS

### 2.1 IN-CONTEXT LEARNING

Brown et al. (2020) demonstrated with GPT-3 the remarkable ability of LLMs to perform a wide range of tasks using only a few examples provided in the input prompt. Few-shot ICL is the ability of large language models to solve new tasks by examining a sequence of (input, label) pairs that share a common concept within the context. Rather than updating their internal parameters, these models rely solely on the contextual examples to deduce the task's rules.

In general, ability to learn from few-shot examples is associated with *meta-learning* (Wang et al., 2020; Hospedales et al., 2021), and success of the ICL demonstrate the strong ability of LLM to meta-learn. In effective ICL, the model infers the underlying task from the examples provided and refines its predictions based on the inferred task. Although this meta-learning-based ability is widely used, the underlying mechanisms enabling LLMs to perform these tasks remain poorly understood, and some puzzling results have been observed. For example, Min et al. (2022b) demonstrated that even when the labels in the examples are randomized, the accuracy improves. Additionally, Chan et al. (2022) have demonstrated that data distributional properties significantly influence ICL performance.

To understand ICL, various approaches have been proposed. For example, Von Oswald et al. (2023); Dai et al. (2023) demonstrated that transformers can solve linear regression problems within the context by leveraging meta-gradients. Based on this, analytical methods have been applied to study the ability of transformers to handle a range of tasks, including discrete functions (Bhattamishra et al., 2023), nonlinear functions (Kim & Suzuki, 2024), and classification problems (von Oswald et al., 2023).

### 2.2 MECHANISTIC INTERPRETABILITY

One promising approach to understanding ICL is *mechanistic interpretability* (MI), which seeks to uncover the internal mechanisms of models (Olah et al., 2020; Elhage et al., 2021; Nanda et al., 2023). A key focus of MI is the study of *circuits*, which are subgraphs with distinct functionality that serve as fundamental building blocks of neural network behavior (Wang et al., 2022; Merullo et al., 2024; Conmy et al., 2023b).

One such circuit studied in the context of ICL is the *induction head* (Olsson et al., 2022). The induction heads are a two-layer structure; in particular, the latter layer is commonly called the induction head, and the earlier layer is referred to as the previous token head. Previous token head attends to and copies the preceding token into the current token. When few-shot examples are present in the context, it chunks each $(x, \ell)$ pair into a single token. Induction heads then perform a match-and-copy operation, matching a query derived from the current token with a key derived from the previous token head's output. For more details on the induction head, see Appendix A. Further research has shown that induction heads can perform soft matching (Crosbie & Shutova, 2024), emerge naturally in multi-head attention settings (Singh et al., 2024), and are present in LLMs (Cho et al., 2024).

Despite these advancements of induction heads, these studies have primarily focused on tasks where the context explicitly includes the label to be copied, such as direct copying tasks. Therefore, induction heads alone cannot fully explain the meta-learning capabilities of ICL in more practical scenarios.

## 3 EXPERIMENTAL SETUP

### 3.1 IN-CONTEXT META LEARNING

To analyze the meta-learning capabilities of ICL, building on prior works (Chan et al., 2022; Reddy, 2023), we design a simple experimental setting named the In-Context Meta-Learning (ICML)

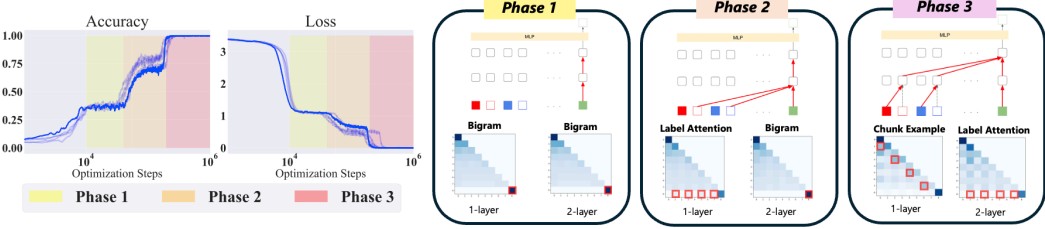

Figure 2: The left panel illustrates the changes in accuracy and loss across three distinct phases during training, with lighter-shaded curves indicating different random seeds. Each phase is highlighted with a different background color: Phase 1 (yellow), Phase 2 (orange), and Phase 3 (red). The right panels visualize the attention maps (circuits) corresponding to each phase, showing the evolution of attention patterns in the two-layer model. Specific attention types, such as Bigram, Label Attention, and Chunk Example, emerge at different phases, reflecting the model's adaptation to the task.

described in Figure 1-(a). Unlike previous approaches, where copying labels or memorizing $(x, \ell)$ pairs was sufficient to predict the answer, our setting instead requires the model to meta-learn the underlying task $(\tau)$ from $(x, \ell)$ context pairs. The network is trained to predict the label of a target $x_q$ given an alternating sequence of $N$ items and $N$ labels:

$$\underbrace{x_1, \ell_1^\tau, x_2, \ell_2^\tau, \ldots, x_N, \ell_N^\tau}_{\text{examples}}, \underbrace{x_q}_{\text{query}}, \underbrace{?}_{\text{prediction}}$$

Here, $\tau$ represents the task, where each task defines a unique $(x, \ell)$ pair with labels $\ell$ randomly assigned to items $x$. The total number of tasks is denoted as $T$, and the context presented to the model consistently corresponds to the same task. Since the query $x_q$ may not be appeared the in-context examples, the network needs to infer the task $\tau$, instead of simply copying a label, from the context.

Following (Reddy, 2023), we represent each item $x$ and label $\ell$ in a $(P + D)$-dimensional space. Of these dimensions, $P$ is dedicated to positional information via a one-hot encoding (with $P = 65$ across all experiments), while $D$ captures the content. To encourage translation-invariant operations, each input sequence is randomly placed within a window of size $(2N + 1)$ spanning the range $[0, P - 1]$. Each class $k$ is associated with a $D$-dimensional mean vector $\mu_k$, whose entries are drawn independently from $\mathcal{N}(0, 1/D)$. For an item $x_i$ assigned to class $k$, we add noise $\eta$ (sampled from the same distribution) scaled by $\epsilon$, giving $x_i = \frac{\mu_k + \epsilon \eta}{\sqrt{1+\epsilon^2}}$, where $\epsilon$ governs within-class variation and the denominator ensures $\|x_i\| \approx 1$. Finally, each class is linked to one of $L$ labels, with $L \leq K$. To control the proportion to which a query can be solved by copying from the context, the same item as the query is included in the context with probability $p_B$. We use $T = 3$, $K = 64$, $L = 32$, $N = 4$, $D = 63$, $\epsilon = 0.1$, $p_B = 0$, unless otherwise specified. In our ICML setup, we can reproduce the standard match-and-copy induction head mechanism from Reddy (2023) by setting $T = 1$, $p_B = 1$,. For detailed results, see Appendix A.

## 3.2 NETWORK STRUCTURE

Following prior research (Reddy, 2023), we use a two-layer attention-only transformer shown in Figure 1-(b), where each layer $\mu$ comprises $m$ heads (indexed by $h$), and a causal mask ensures position $i$ attends only to positions $j \leq i$. A two-layer MLP classifier then produces the label probabilities. For the complete set of equations and hyperparameter details, see Appendix B. In this architecture, each head $h$ in layer $\mu$ computes attention weights $\{p_{ij}^{(\mu,h)}\}$, quantifying how strongly position $i$ (query) attends to position $j$ (key). These outputs are aggregated across heads and passed to the MLP, which makes the final label predictions.

The classifier is a two-layer MLP with ReLU activations, followed by a softmax layer producing probabilities over $L$ labels. We train this network to classify the query item $x_q$ into one of the $L$ labels using cross-entropy loss. Both the query/key dimension and the MLP hidden layer dimension are set to 128. We use a batch size of 128 and optimize with vanilla stochastic gradient descent at a learning rate of 0.01.

# 4 PHASE TRANSITIONS AND EMERGENT CIRCUITS

## 4.1 THREE-PHASE DYNAMICS AND CIRCUIT OVERVIEW

We conducted experiments under the ICML setting with three tasks (i.e., $T = 3$). As shown on the left side of Figure 2, the results reveal three distinct phases of accuracy changes, each accompanied by a corresponding drop in loss. The observed dynamics are as follows: the first accuracy plateau occurs at around 30–40%, the second at approximately 75%, and the final phase reaches 100%. To clearly these three phases, we define the following metric:

$$\Delta\text{Accuracy} = \text{Accuracy}(t + \Delta t) - \text{Accuracy}(t),$$

where $t$ denotes the optimization step and we set $\Delta t = 100$. In Figure 3, we plot this quantity along with the model's accuracy, marking vertical lines at steps where $\Delta\text{Accuracy} > 0.025$. These lines serve as boundaries between the three observed phases. Based on this threshold, we partition the model's behavior into Phase 1, Phase 2, and Phase 3 throughout the remainder of this paper.

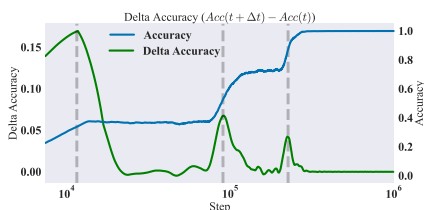

Figure 3: Accuracy (blue) and $\Delta$Accuracy (green) as functions of the training step. Here, $\Delta\text{Accuracy} = \text{Acc}(t + \Delta t) - \text{Acc}(t)$ with $\Delta t = 100$. Vertical dashed lines indicate where $\Delta$Accuracy exceeds 0.025, marking the transition points between the three observed phases (Phase 1, Phase 2, Phase 3

On the right side of Figure 2, we visualize the attention maps from the two layers of the model during each phase. The attention patterns emerging during the learning process can be categorized into the following three types:

1. **Bigram:** Strong attention is focused on the token in the context that corresponds to the query token ($x_q$).
2. **Label Attention:** Strong attention is focused on the label tokens of the $(x, \ell)$ pair within the context.
3. **Chunk Example:** Attention aggregates the $(x, \ell)$ token pair in the context into a single token, similar to the induction head's previous token head.

As visualized on the right side of Figure 2, the combinations of these attention types differ between the first and second layers across the three phases:

**Phase 1 (Non-Context Circuit; NCC):** Both layers use bigram attention, *ignoring the context* and relying solely on the model's weights. At this stage, the model predict label base on only query, limiting accuracy to around $1/T$. In this case, with three tasks, the accuracy stagnates at around 30–40%.

**Phase 2 (Semi-Context Circuit; SCC):** The first layer exhibits label attention, while the second layer focuses on the query token (bigram attention). The model not only leverages weights memory but also attends to label tokens (i.e., *half of the context*), in the context to infer possible answers, resulting in improved accuracy of approximately 75%. We delve into these details in subsection 4.3

**Phase 3 (Full-Context Circuit; FCC):** The first layer aggregates the $(x, \ell)$ pair into a single token (chunk example), while the second layer focuses on these aggregated tokens (label attention) to predict label, resulting in *using the entire context*. Through this abstraction of the pairwise relationship (i.e., task inference), the model can produce correct answers for the query. Once the model learns this circuit, it achieves 100% accuracy.

The relationship between each circuit and its corresponding attention pattern is summarized in Table 1.

## 4.2 QUANTIFYING CIRCUIT EMERGENCE

To quantitatively measure these circuits, we propose three metrics based on the attention maps of each layer. Let $p_{i,j}^{\mu,h}$ represent the attention from token $j$ to token $i$ in the $h$-th head of the $\mu$-th layer. Let the context length be $2N + 1$ (in this case, $N = 4$). We define three primary attention-based metrics, with precise formulas provided in Table 2. Here, we briefly describe what each metric represents: (1) *Bigram Metrics* capture the attention from the query token to itself; (2) *Label Attention Metrics*

Table 1: Summary of circuits, accuracy, and layer-wise attention.

| Circuit | Accuracy ($T = 3$) | Layer 1 | Layer 2 |
|---------|-------------------|---------|---------|
| NCC | 30–40% | Bigram | Bigram |
| SCC | ≈75% | Label Attention | Bigram |
| FCC | 100% | Chunk Example | Label Attention |

Table 2: Formulas of the three attention metrics.

| Metric | Formula |
|--------|---------|
| Bigram | $p_{2N+1,2N+1}^{\mu,h}$ |
| Label Attention | $\sum_{k=1}^{N} p_{2N+1,2k}^{\mu,h}$ |
| Chunk Example | $\frac{1}{N}\sum_{k=1}^{N} p_{2k,2k-1}^{\mu,h}$ |

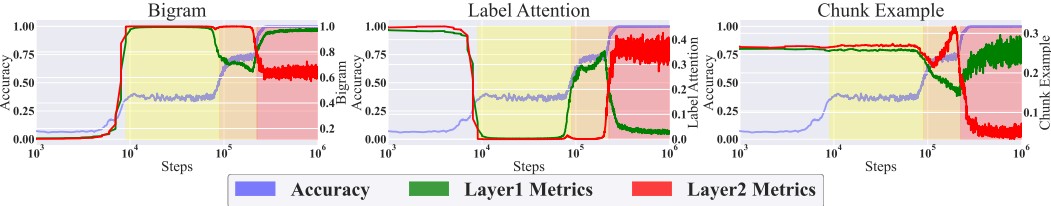

Figure 4: Evolution of the three attention metrics (Bigram, Label Attention, and Chunk Example) across optimization steps for the first (green) and second (red) layers. The shaded regions represent the three learning phases: Phase 1 (yellow), Phase 2 (orange), and Phase 3 (red), defined by ΔAccuracy (Figure 3). Each metric shifts cleanly at the phase boundaries, demonstrating a close correspondence between accuracy improvements and circuit-level transformations.

measure the total attention from the query token to the label tokens within the context; (3) *Chunk Example Metrics* assess the attention from $x$ to $\ell$ within each $(x, \ell)$ pair.

The plots in Figure 4 illustrate how these metrics evolve in the first and second layers across the three phases. For the Bigram Metrics, both the first and second layers show high values at the moment of the initial jump in accuracy, marking the formation of the NCC. Then, at the beginning of Phase 2, the bigram metrics in the first layer decrease significantly while those in the second layer remain high, and label attention in the first layer rises — together leading to the formation of the SCC. At the start of Phase 3, the chunk example metrics in the first layer increase, and the label attention metrics in the second layer also become high, resulting in the formation of the FCC. Importantly, these metric transitions align closely with the corresponding jumps in model accuracy, supporting the view that these metrics provide a valid and quantitative perspective on the circuit changes observed during the three phases, as depicted in the right side of Figure 2.

### 4.3 DEEPER LOOK AT THE SEMI-CONTEXT CIRCUIT

**How SCC Drives Accuracy Gains** In Phase 2, the model forms the SCC, using label information from the context in addition to the query input ($x_q$). We provide a theoretical analysis of why this leads to improved accuracy and empirically validate our theory through controlled experiments. To clarify SCC's behavior, we tested the following simplified conditions:

1. The number of classes ($K$) equals the number of labels ($L$), with no duplication.

2. The input context (including the query) contains no duplicate classes.

3. The number of tasks ($T$) is set to 2.

4. There are no common $(x, \ell)$ pairs shared across tasks.

5. To specifically focus on SCC, a mask is applied to circuits associated with SCC during training (details are provided in the Appendix C).

In Phase 1, since there are two tasks, the model has a 50% chance of predicting correctly by random guessing. In other words, the model's prediction reduces to a binary choice for each input query ($x_q$).

Once label information becomes usable, the binary choice can potentially be narrowed further. This occurs when one of the labels corresponding to the two options is present in the context. In this scenario, the label in the context is definitively not the correct answer for the query, as per the defined conditions. Thus, the answer becomes uniquely determinable, increasing accuracy. Following the

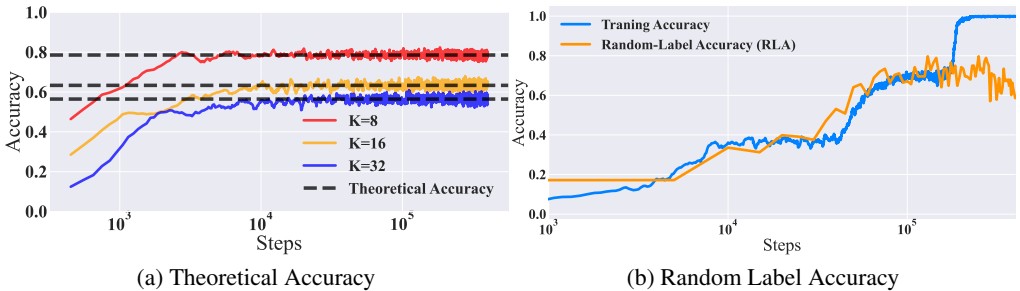

Figure 5: (a) Comparison of theoretical accuracy (dashed lines) and model accuracy for different class counts ($K$). The close alignment between theoretical predictions and experimental results confirms the validity of the theoretical analysis. (b) Comparison of training accuracy and random-label accuracy (RLA). The plot demonstrates the rise in both metrics, with RLA following a trend similar to emergence Phase 2. This indicates that SCC acquired in Phase 2 contributes to improved accuracy even with shuffled labels.

derivation in Appendix D, the probability of one of the labels appearing in the context is

$$p = 1 - \frac{\binom{K-2}{4}}{\binom{K-1}{4}}.$$

Therefore, the theoretical accuracy achievable with SCC can be expressed as:

$$\texttt{Theoretical Accuracy} = p \cdot 1 + (1 - p) \cdot 0.5.$$

Figure 5-(a) shows the theoretical accuracy alongside the accuracy achieved by a model trained with only Phase 2 attention circuits remaining. The class/label counts were varied as $K = \{8, 16, 32\}$. The near-perfect agreement between the theoretical and empirical results confirms both the validity of our derivation and the role of SCC in boosting accuracy.

**Random-Label Robustness of SCC**    We focus on the tendency of SCC to make predictions "based solely on labels and query." We hypothesize that this circuit explains the puzzling phenomenon that the improvement in ICL performance observed even when using random labels, as noted in Min et al. (2022b). Min et al. (2022b) has demonstrated that replacing labels randomly within examples results in only a marginal performance drop, suggesting that ICL does not rely heavily on $(x, \ell)$ pairs. To investigate this phenomenon, we define an Out-of-Distribution (OOD) evaluation where the labels in each context pair are randomly permuted. Specifically, we consider:

$$\underbrace{x_1, \ell^\tau_{\pi(1)}, x_2, \ell^\tau_{\pi(2)}, \ldots, x_N, \ell^\tau_{\pi(N)}}_{\text{examples}}, \underbrace{x_q,}_{\text{query}} \underbrace{?}_{\text{prediction}}$$

Here, $\pi$ is a random permutation on $\{1, 2, \ldots, N\}$, meaning that $\ell^\tau_{\pi(i)}$ replaces the original label $\ell^\tau_i$. By measuring the model's accuracy under these shuffled labels, we obtain the *Random-Label-Accuracy (RLA)*. In the Figure 5-(b), we compare this RLA with the training accuracy. Similar to the rise observed in Phase 2, when SCC is acquired, the RLA also increases. This suggests that the reason for the improved performance with random labels, as seen in Min et al. (2022b), is the existence of circuits similar to SCC within LLMs.

## 4.4    EFFECTS OF DATA PROPERTY ON CIRCUITS EMERGENCE

Previous studies have indicated that certain properties of the training data, such as burstiness, can influence the emergence of ICL (Chan et al., 2022) and induction heads (Reddy, 2023). In this work, we explore how these data properties affect the development of circuits in our ICML setting, with the aim of advancing our understanding of the multi-phase emergence of these circuits. As mentioned in section 3, the variables capturing the characteristics of the data include the number of tasks $T$, the number of classes $K$, the noise magnitude $\epsilon$. In addition, and following Chan et al. (2022), we adopt rank-frequency distributions over both classes and tasks: $f(k) \sim k^{-\alpha}$ and $f(\tau) \sim \tau^{-\beta}$, which follow a power-law form commonly known as *Zipf's law* Zipf (1949) (see Appendix E for details).

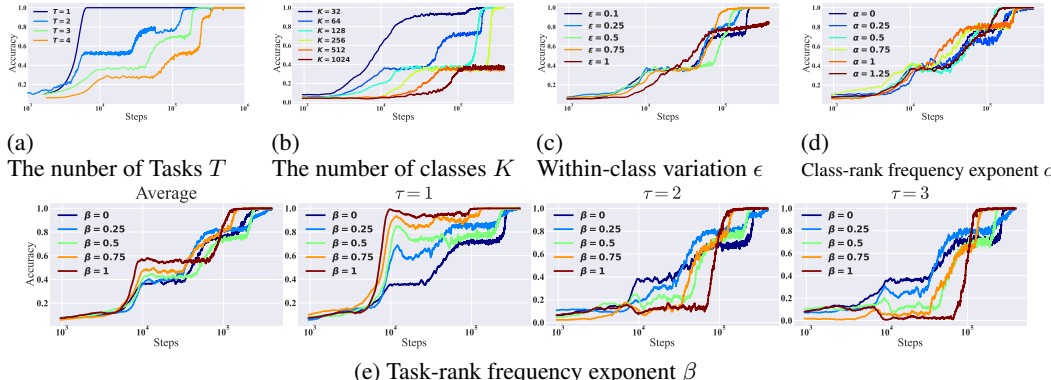

Figure 6: The relationship between multi-phase transitions and data distribution properties is explored by varying key parameters: the number of tasks ($T$), the number of classes ($K$), the noise magnitude ($\epsilon$), the sampling bias for classes ($k^{-\alpha}$), and the sampling bias for tasks ($\tau^{-\beta}$). Default values are $T = 3$, $K = 64$, $\epsilon = 0.1$, $\alpha = 0$, and $\beta = 0$. The plots show how these variations influence accuracy and the emergence of phase transitions.

The default values are $T = 3$, $K = 64$, $\epsilon = 0.1$, $\alpha = 0$, and $\beta = 0$. The results of varying these parameters are shown in the Figure 6. For results obtained by varying $p_B$, see Appendix F.

In Figure 6-(a), we present the results of varying the number of tasks $T$. As $T$ increases, Phase 1 accuracy decreases (approximately proportional to $1/T$). When $T = 1$, the setup aligns with previous studies (see Figure 1), where the model's accuracy increases in a single phase rather than undergoing multiple phases. Conversely, for $T \geq 2$, the model consistently exhibits three distinct phases. This indicates that the multi-phase phenomenon is robust to the number of tasks, and that introducing additional tasks in the ICL setting can provide new empirical insights.

In Figure 6-(b), when $K$ is small (e.g., $K = 32$), the model tends to skip Phase 1 and transition directly to Phase 2. In contrast, when $K$ is large (e.g., $K = 128, 256$), the model skips Phase 2 and jumps directly from Phase 1 to Phase 3. This can be explained by the theoretical values derived in subsection 4.3, where increasing the number of classes brings the accuracy in Phase 2 closer to that in Phase 1, effectively making Phase 2 unobservable for large $K$.

In Figure 6-(c), increasing $\epsilon$ (the within-class variation) leads to skipping Phase 2. Moreover, when $\epsilon$ is 1, Phase 1 is also skipped. Following the results of Chan et al. (2022), higher values of $\epsilon$ make it more difficult for the model to memorize the $(x, \ell)$ pairs in its weights, and thus it shifts its focus toward leveraging the context. The observation that NCC is skipped entirely when $\epsilon = 1$ aligns with this trend. Although SCC is a circuit that uses the context, it inherits the nature of NCC, causing it to be skipped as $\epsilon$ increases. In Figure 6-(d), we see that increasing $\alpha$ likewise tends to skip Phase 1 or Phase 2. The heightened sampling bias makes it more challenging to memorize pairs in the weights, so the model more readily exploits context-based information. As a result, the NCC or SCC does not emerge. In summary, the results suggest that when the model finds it difficult to memorize $(x, \ell)$ pairs (larger $\epsilon$ or $\alpha$) neither NCC nor SCC emerges.

In Figure 6-(e), we examine how varying the task sampling bias $\beta$ affects both the average accuracy across tasks and the accuracy of each individual task. While changing $\beta$ leads to only minor differences in the overall average trend, the accuracy on a per-task basis varies considerably with $\beta$. In particular, when $\beta$ is high (e.g., $\beta = 1$), the model tends to memorize the most frequent task (i.e., $\tau = 0$) first, causing the remaining tasks to skip NCC and progress directly to forming FCC.

# 5 MULTI-HEAD ENHANCES CIRCUIT DISCOVERY

## 5.1 PARALLEL CIRCUIT EXPLORATION

To investigate a more practical scenario, we extend our analysis to multi-head attention. Figure 7-(a) compares the accuracy changes for models with two heads and one head. In the left panel of Figure 7-(a), we observe that phase transitions become less pronounced when using multi-head attention. A closer examination of the attention maps for each head (as shown in the right panel of Figure 7-(a)) reveals that different heads specialize in distinct functions. Specifically, one head learns circuits

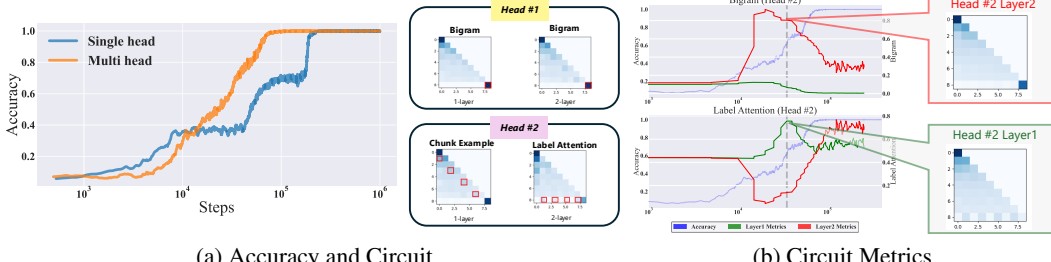

(a) Accuracy and Circuit              (b) Circuit Metrics

Figure 7: (a) Comparison of accuracy dynamics between single-head (blue) and multi-head (orange) attention models (left). The multi-head model exhibits smoother accuracy improvements, without the distinct phase transitions observed in the single-head model. On the right, the attention maps for the two heads in the multi-head model are visualized. Head 1 specializes in NCC, while Head 2 adopts circuits resembling FCC. These findings indicate that multi-head attention allows parallel circuit discovery, enhancing the efficiency of the learning process. (b) Circuit metrics (left) and attention maps (right) for Bigram (Head 2) and Label Attention (Head 2) in multi head setting. The left plots depict the progression of Accuracy (blue), Layer 1 Metrics (green), and Layer 2 Metrics (red) over training steps. The attention maps on the right correspond to the model's behavior at 30,000 training steps, as indicated by the vertical dashed line.

resembling NCC, while another head becomes FCC. This parallel specialization provides a smoother trajectory of accuracy improvement, in contrast to the multi-phase transitions observed in single-head models.

These findings suggest that multi-head attention allows for parallel exploration of circuits, improving the efficiency of circuit discovery. As a result, the multiple phase transitions characteristic of single-head models are absent in multi-head configurations. This behavior aligns with observations in LLMs, where multi-head attention enables different heads to serve distinct functions, leading to smoother accuracy improvements, as seen in Figure 7. Results for a larger number of heads are provided in the Appendix G.

## 5.2 HIDDEN CIRCUIT EMERGENCE

In Figure 7-(b), we observe multiple attention heads lead to smoothing the accuracy improvement. To gain deeper insights into this phenomenon, we analyze how the internal circuits evolve by using the circuit metrics summarized in Table 2. In Figure 7-(b) (left), we present the circuit metrics for Bigram and Label Attention in Head 2. Notably, around the 30,000th training step, the Bigram metric exhibits a pronounced increase in the second layer, whereas the Label Attention metric is notably larger in the first layer. The right panel displays the corresponding attention maps, which clearly demonstrate an SCC-like pattern, illustrating how the model's attention shifts between bigram-driven and label-focused mechanisms. The attention maps on the Figure 7-(b) correspond to the model's behavior at 30,000 training steps, as indicated by the vertical dashed line. A complete set of metrics is provided in Appendix H.

These results suggest that, even though we do not observe multiple-phase transitions in accuracy under the multi-head configuration, a *hidden circuit* emerge within the model's internal mechanisms. This hidden phenomenon implies that, in more practical scenarios (such as large-scale language model where the loss typically decreases in a smooth fashion), the model's internal circuits may still undergo significant emergent shifts.

## 6 DISCUSSION

We introduced controlled experimental called In-Context Meta-Learning (ICML), designed to move beyond simple copy tasks by requiring task inference. We then investigate how a 2-layer, attention-only transformer acquires ICL abilities, inspired by induction head research (Olsson et al., 2022; Reddy, 2023). Although our model is much smaller than those used in large-scale interpretability research (Wang et al., 2022; Merullo et al., 2024; Templeton et al., 2024; Gao et al., 2024), this controlled design revealed novel insights, including multi-phase transitions that illuminate how the

model's internal circuits evolve. Moreover, the observed random-label robustness (subsection 4.3) and multi-head behaviors, where the loss decreases smoothly (section 5), both align with findings in LLMs. These results connect small-scale experiments to practical LLMs, clarifying ICL mechanisms.

**Relationship to Prior Internal-Circuit Research**    Previous investigations taking an internal-circuit approach to ICL have largely focused on induction heads, which employ a match-and-copy mechanism (Ren et al., 2024; Cho et al., 2024). In contrast, by adopting a more practical meta-learning perspective, our study reveals multi-phase circuits that initially memorize examples and then evolve to infer the underlying task, which differs from the single-phase transitions commonly observed in induction heads. While both induction heads and our Full-Context Circuits (FCC) chunk contextual $(x, \ell)$ pairs into a single token in the first layer, the second layer diverges: induction heads retrieve only a label, whereas FCC further aggregates (Chunk Example $\rightarrow$ Label Attention in Table 1). This shared mechanism in the first layer implies that even a simple copy task contributes to meta-learning–like ICL capabilities. In addition, consistent with earlier findings (Chan et al., 2022; Singh et al., 2023; Reddy, 2023), these results highlight the key role of dataset characteristics in circuit formation and ICL

**Implication for LLMs**    Our analysis links circuits to the established concept of task vectors (Hendel et al., 2023; Todd et al., 2024). A task vector represents the abstracted representation a model forms from examples, and although such vectors have been recognized, the internal circuit-based mechanisms that produce them remain poorly understood. Our findings offers a step toward elucidating these mechanisms. In addition, we examine multi-head attentions. Prior work (Singh et al., 2024) has identified redundancy in induction heads under multi-head architectures. Our findings indicate that, rather than mere redundancy, multiple distinct circuits emerge in parallel in the multi-head setting, resulting in smoother performance gains. This observation bridges the discontinuous concept of circuits with the continuous performance improvements seen in LLMs.

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

## A   INDUCTION HEAD

Figure 8 illustrates an induction circuit consisting of a *previous token head* in Layer 1 and an *induction head* in Layer 2. After Layer 1, the side-by-side $x$ and $\ell$ tokens are chunked into a single token. In Layer 2, two operations occur: **matching** of $x$ via queries and keys (in purple) and **copying** of $\ell$ (in red).

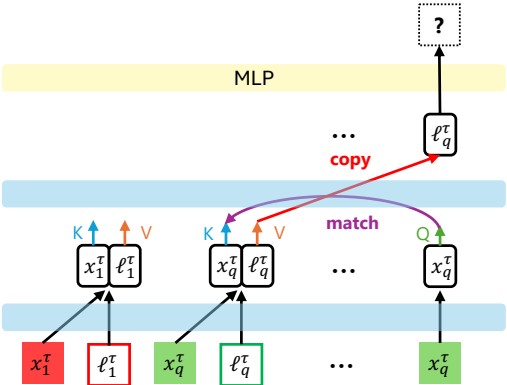

Figure 8: The circuit consists of a previous token head in Layer 1 and an induction head in Layer 2. After Layer1, the side-by-side $x$ and $\ell$ tokens are chunked into a single token. In Layer 2, we highlight two operations: **matching** of $x$ vai queries and keys (in purple), and **copying** of $\ell$ (in red).

When a sample is drawn with probability $p_B$, the burstiness parameter $B$ introduced by Chan et al. (2022); Reddy (2023) becomes relevant, determining how many times items from the query class appear in an input sequence (where $N$ is a multiple of $B$). In our ICML setup, we specifically examine the case with $T = 1$, $p_B = 1$, and $B = 1$, as shown in Figure 9. We observe that the first attention layer encodes each $(x, \ell)$ pair into a single token, while the second layer strongly attends to one of these pairs, effectively implementing the match-and-copy mechanism characteristic of an induction head. Notably, our setting thus subsumes the standard induction head experiments proposed in Reddy (2023).

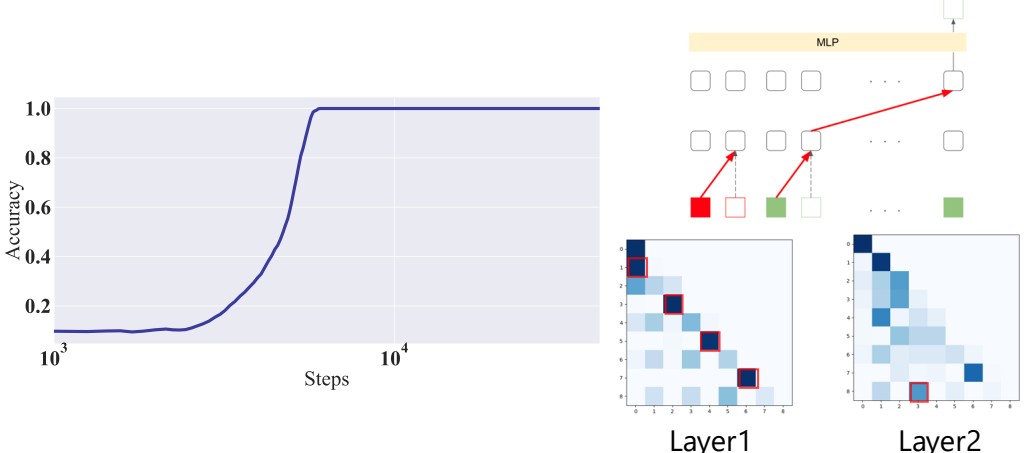

Figure 9: (left) The emergence of induction heads is observed as single-phase transition. (right) The attention maps on the right illustrate the circuit mechanism, where Layer 1 groups $(x, \ell)$ pairs into single-token representations, and Layer 2 then copies this label.

# B MODEL DETAILS

## B.1 NETWORK ARCHITECTURE

Our model features two layers of **multi-head attention** with a causal mask, followed by a two-layer MLP classifier. Each attention layer $\mu \in \{1, 2\}$ has $m$ heads, labeled by $h$. Let $(u_1, \ldots, u_n)$ be the input sequence (subject to a causal mask ensuring $i$ can only attend to $j \leq i$). The outputs of the first layer are $\{v_i\}$; those of the second layer are $\{w_i\}$.

**Attention Computation.** Within layer $\mu$, head $h$ computes attention weights

$$p_{ij}^{(\mu,h)} = \frac{\exp\left(\left(K_\mu^{(h)} u_j\right)^\top \left(Q_\mu^{(h)} u_i\right)\right)}{\sum_{k \leq i} \exp\left(\left(K_\mu^{(h)} u_k\right)^\top \left(Q_\mu^{(h)} u_i\right)\right)}, \tag{1}$$

where $Q_\mu^{(h)}$ and $K_\mu^{(h)}$ are the learnable query and key matrices for head $h$ in layer $\mu$. Next, each head outputs a weighted sum of the *value*-transformed inputs:

$$\text{Head}_i^{(\mu,h)} = \sum_{j \leq i} p_{ij}^{(\mu,h)} \left(V_\mu^{(h)} u_j\right), \tag{2}$$

where $V_\mu^{(h)}$ is the corresponding value matrix.

**Multi-Head Aggregation.** The outputs of all $m$ heads in layer $\mu$ are concatenated and projected by a trainable matrix $W_{O\mu}$, yielding

$$v_i = u_i + W_O^1 \left[\text{Head}_i^{(1,1)}; \ldots; \text{Head}_i^{(1,m)}\right], \tag{3}$$

$$w_i = v_i + W_O^2 \left[\text{Head}_i^{(2,1)}; \ldots; \text{Head}_i^{(2,m)}\right]. \tag{4}$$

Here, $[\ldots]$ indicates concatenation over the head outputs, and each $W_{O\mu}$ is a learnable linear projection.

**Classifier.** The two-layer MLP receives the final attention outputs $\{w_i\}$ (e.g., specifically $w_n$, if $n$ indexes the query token). A hidden layer with ReLU activation is followed by a softmax that produces label probabilities.

## B.2 TRAINING DETAILS

Table 3: Training and Model Configuration

| Hyperparameter | Value |
|---|---|
| Loss Function | Cross-entropy |
| Optimizer | Vanilla SGD |
| Learning Rate | 0.01 |
| Batch Size | 128 |
| Dimension of query/key/value | 128 |
| MLP Hidden Layer Dimension | 128 |
| Causal Mask | Restrict sums to $j \leq i$ |

## C    CONTROLLED CIRCUIT PRUNING EXPERIMENTS

To validate the relationship between the identified circuits and model performance, we conducted controlled pruning experiments. In these experiments, all components except the circuits corresponding to a specific phase were pruned at initialization, isolating the contribution of each circuit. For comparison, we also trained a fully trainable model, referred to as the *full model*, which could attend to all identified attention patterns.

As shown in Figure 10, networks trained with only the circuits from a particular phase plateaued at accuracies corresponding to that phase. This result provides strong evidence that the circuits identified in each phase are directly responsible for the observed performance.

Interestingly, when the Phase 3 circuit was provided from the beginning (pink curve in Figure 10), the model achieved 100% accuracy in single step. In contrast, the full model exhibited a more gradual improvement, sequentially discovering and leveraging the circuits corresponding to each phase. This highlights the dynamic nature of the full model's training process, where it incrementally constructs and refines the required circuits during training.

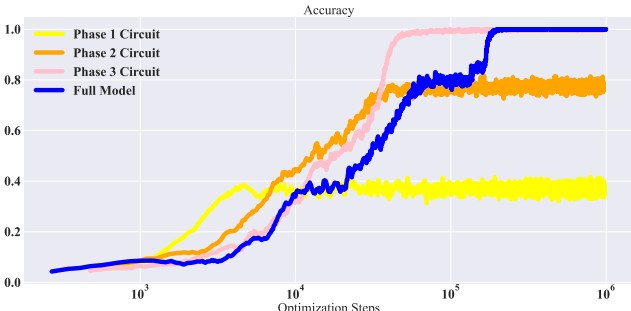

Figure 10: Controlled pruning experiments to validate the relationship between identified circuits and model performance. Networks trained with only the circuits from a specific phase plateaued at accuracies corresponding to that phase (yellow: Phase 1, orange: Phase 2, pink: Phase 3). This demonstrates that the identified circuits are directly responsible for the observed performance in each phase.

## D    DERIVATION OF THE THEORETICAL ACCURACY

In the main text, we define

$$p = 1 - \frac{\binom{K-2}{4}}{\binom{K-1}{4}}, \tag{5}$$

and use it to obtain the "Theoretical Accuracy" as

$$\texttt{Theoretical Accuracy} = p \cdot 1 + (1-p) \cdot 0.5. \tag{6}$$

This appendix provides a more detailed derivation of these formulas, along with the underlying conditions.

**Task Conditions.**

1. The number of classes ($K$) equals the number of labels ($L$), with no duplication.

2. The input context (including the query) contains no duplicate classes.

3. Only two tasks are considered ($T = 2$).

4. There are no common $(x, \ell)$ pairs shared between different tasks.

5. To focus on SCC, a mask is applied to circuits associated with SCC during training.

**Excluding Both $L_1$ and $L_2$.** We are interested in the probability that the context does *not* contain $L_1$ or $L_2$.

- The total number of ways to choose 4 distinct classes from the $K-1$ classes (excluding the query's class) is $\binom{K-1}{4}$.
- To exclude both $L_1$ and $L_2$, we must choose all 4 classes from the remaining $K-2$ classes, leading to $\binom{K-2}{4}$ possible ways to form the context with neither $L_1$ nor $L_2$ present.

Hence, the probability that *neither* $L_1$ nor $L_2$ is in the context is

$$\frac{\binom{K-2}{4}}{\binom{K-1}{4}}.$$

**Probability $p$ and the Accuracy Calculation.** We denote by $p$ the probability that *at least one* of $L_1$ or $L_2$ appears in the context:

$$p = 1 - \frac{\binom{K-2}{4}}{\binom{K-1}{4}}.$$

Under the task rules, if at least one of these two labels appears in the context, it *cannot* be the label for the query, so the other one must be correct. This yields 100% accuracy in that scenario. Conversely, if *neither* $L_1$ nor $L_2$ is found in the context (probability $1-p$), the model is forced to guess between two equally likely options, resulting in 50% accuracy. Therefore,

$$\texttt{Theoretical Accuracy} = p \cdot 1 + (1-p) \cdot 0.5,$$

as stated in the main text.

# E   RANK-FREQUENCY

In natural language processing and other real-world domains, both data instances and task distributions often follow a power-law structure, commonly referred to as *Zipf's law* (Zipf, 1949). This law states that the frequency of an item or task is inversely proportional to its rank, meaning that a small number of elements occur frequently, while the majority appear rarely. Formally, this is expressed as:

$$f(k) \propto k^{-\alpha}, \tag{7}$$

where $k$ denotes the rank of an item, and $\alpha$ controls the degree of skewness. Figure 11 illustrates how increasing $\alpha$ leads to a more imbalanced distribution, with a steep drop in frequency beyond the highest-ranked elements.

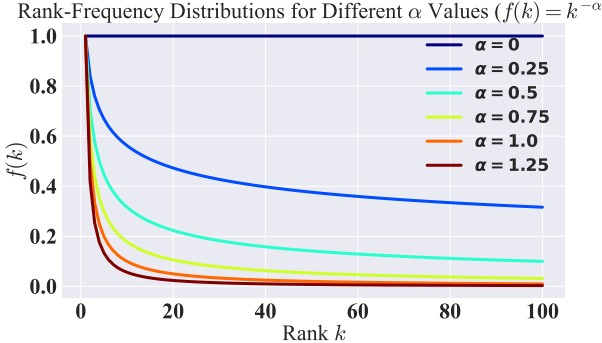

Figure 11: Rank-frequency distributions for different values of the power-law exponent $\alpha$, following the Zipfian distribution $f(k) = k^{-\alpha}$. As $\alpha$ increases, the distribution becomes more skewed, with a few high-frequency items dominating while the majority appear infrequently.

In our setting, not only data but also task sampling follows a similar Zipfian distribution:

$$f(\tau) \sim \tau^{-\beta}, \tag{8}$$

where $\beta$ determines the skewness of the task distribution.

## F    EFFECTS OF BIRSTINESS ON CIRCUIT EMERGENCE

When a sample is drawn with probability $p_B$, the burstiness parameter $B$ introduced by Chan et al. (2022); Reddy (2023) becomes relevant, determining how many times items from the query class appear in an input sequence (where $N$ is a multiple of $B$). Figure 12 examines the impact of burstiness $B$ and probability $p_B$. The left panel shows accuracy curves for different values of $B$ at a fixed $p_B = 0.25$. As $B$ increases, Phase 1, where NCC memorizes pairs through weight updates — tends to be skipped. The right panel presents accuracy curves for different values of $p_B$ while keeping $B = 1$ fixed. As $p_B$ increases, the model's accuracy improves more smoothly, and distinct phase transitions become less pronounced. These results align with previous studies showing that increased burstiness tends to shift the model away from weight-based solutions and toward context-dependent reasoning (Chan et al., 2022; Reddy, 2023).

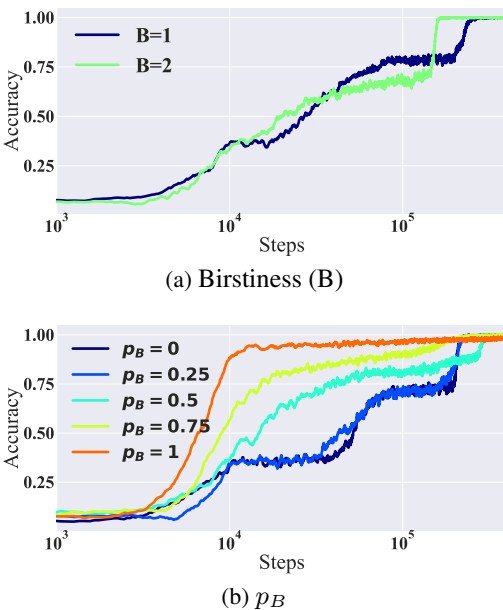

(a) Birstiness (B)

(b) $p_B$

Figure 12: **(Left)** Accuracy curves for different values of $B$ at a fixed $p_B = 0.25$. Increasing $B$ tends to skip Phase 1, where NCC memorizes pairs through weights. **(Right)** Accuracy curves for different values of $p_B$ with $B = 1$. As $p_B$ increases, the learning process becomes smoother, reducing the occurrence of distinct phase transitions.

## G    MULTIHEADS EXPERIMENTS

Figure 13 (Left) shows accuracy curves over training steps for different numbers of attention heads (1, 2, 4, 8, and 16). Models with multiple heads exhibit a smooth increase in accuracy, whereas the single-head configuration undergoes multi-phase transitions, where accuracy improves in distinct jumps rather than gradually.

Figure 13 (Right) visualizes attention patterns in a 4-head attention model across two layers. The four heads naturally divide into two functional roles: two heads focus on NCC, while the other two heads focus on FCC.

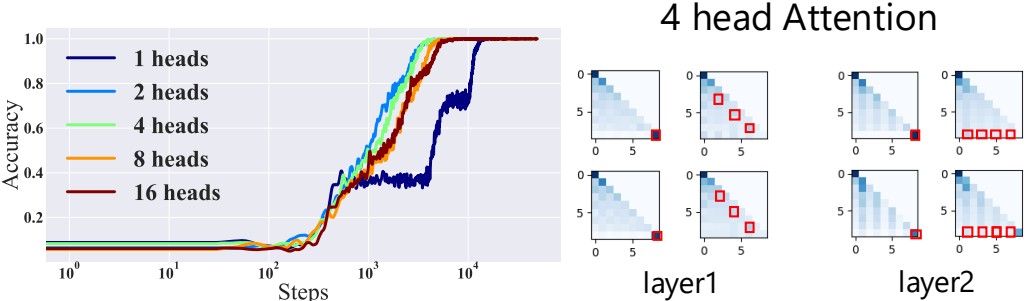

Figure 13: (Left) Accuracy curves over training steps for different numbers of attention heads (1, 2, 4, 8, and 16). Models with multiple heads exhibit a smooth increase in accuracy, whereas the single-head configuration shows multi-phase transition. (Right) Visualization of attention patterns in a 4-head attention model, separated by layer. Two heads focus on NCC, while the other two focus on FCC. Red squares highlight key attention positions indicative of each role.

# H  CIRCUIT METRICS IN MULTIHEAD ATTENTION

Figure 14 presents circuit metrics for each attention head, analyzed by layer in a two-head attention model. Head 1 consistently maintains high bigram values across both Layer 1 and Layer 2. This indicates that it primarily performs token-level copying operations, forming an NCC. In contrast, Head 2 exhibits a different pattern. As training progresses, the chunk example metric increases in Layer 1, while the label attention metric becomes dominant in Layer 2, forming an FCC.

These findings reinforce the idea that multi-head attention facilitates specialization, allowing different heads to develop distinct computational circuits that enhance the model's meta-learning capabilities.

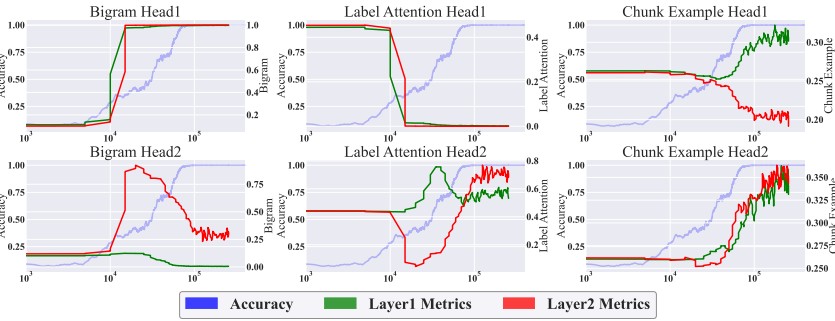

Figure 14: Circuit metrics for each attention head, analyzed by layer in two heads attentions. Head 1 maintains high bigram values across both Layer 1 and Layer 2, indicating the formation of an NCC. In contrast, Head 2 exhibits increasing chunk example values in Layer 1 and high label attention values in Layer 2, suggesting the formation of an FCC.

