# OpenReview forum: "In-Context Meta Learning Induces Multi-Phase Circuit Emergence"
_ICLR.cc/2025/Workshop/BuildingTrust — BuildingTrust_

### Official Review · Reviewer_gYeJ · 2025-02-24
**Excellent Contribution to Mechanistic Interpretability**

**Rating:** 9
**Confidence:** 4

**Review:**

This paper offers a precise and intelligent method for probing the circuit phase changes that drive model ICL capabilities. The challenge they pose, task inference, is much more meaningful than simple copy tasks and therefore provides meaningful insights for model internal workings. The work is done with an appropriate level of rigor – the circuit emergence metrics in Figure 4, theoretical accuracy guarantees in  Figure 5, and parameter sweeps in Figure 6 are informative and add high credibility to the results and their interpretations. I believe this paper is well executed and instructive on how to perform rigorous circuit analysis in language models.
One experiment that could be interesting to see and easy to implement is an exploration of how scaling model size impacts the phase change trajectories. Increasing model size will certainly change the memorization capacity, likely changing the accuracy trajectories for all of the sweeps in Figure 6. It would also be interesting to see how larger attention layers affect the ability of the model to learn full and semi-context circuits, a result I think is less clear and potentially interesting. However, even without this experiment, I still think that this paper is well-scoped and an excellent contribution.
One final note is that I think there might be merit for the discussion of task vectors to be moved out of the obscurity of the appendix and into the paper. This implication seems important and potentially the inspiration for future work, and therefore I think it should be displayed more prominently.

---

### Official Review · Reviewer_4XwW · 2025-03-02
**Review for submission 38**

**Rating:** 7
**Confidence:** 4

**Review:**

This study explores the training dynamics of in-context meta-learning tasks, shedding light on the emergence of multi-phase circuits throughout the training process. Initially, the research uncovers bigram-type circuits that concentrate solely on the query. Subsequently, a circuit emerges in the second phase that focuses solely on the labels within the context. Finally, in the last phase, a circuit forms that chunks each example pair into a single token.

Overall, this paper uncovers a compelling pattern in LLM training, illustrating how circuits develop at various stages. One limitation is that the study centers on a single task. It may be valuable to broaden the scope by considering a variety of tasks with differing complexities (see [1], which covers a similar task in this paper). By examining the phases of circuit emergence across different tasks and comparing them, a deeper understanding of the relationship between circuit emergence ability and task difficulty could be achieved.

[1] Chen and Zou, What Can Transformer Learn with Varying Depth? Case Studies on Sequence Learning Tasks, ICML 2024

---

### Official Review · Reviewer_jA8u · 2025-03-02
**This paper investigates how transformers acquire meta-learning capabilities through in-context learning (ICL).**

**Rating:** 8
**Confidence:** 4

**Review:**

### **Summary**
This paper investigates how transformers acquire meta-learning capabilities through in-context learning (ICL). By extending the copy task to an In-Context Meta-Learning (ICML) setup, the authors identify three distinct phases of circuit emergence (NCC, SCC, FCC) that enable task inference. Their analysis reveals how data properties and multi-head attention influence circuit dynamics, bridging mechanistic insights to practical LLM behaviors.

### **Pros:**
1. **Novelty and Significance**:
   - Identifies **multi-phase circuit emergence** during meta-learning, a novel contribution beyond prior work on induction heads.
   - Links circuit dynamics to practical phenomena (e.g., random-label robustness in LLMs), enhancing relevance to real-world ICL.
   - Introduces **quantitative metrics** (Bigram, Label Attention, Chunk Example) to systematically track circuit evolution.

2. **Methodological Rigor**:
   - Controlled experiments with a simplified transformer (2-layer architecture) enable clear isolation of circuit behaviors.
   - Validates theoretical predictions (e.g., Phase 2 accuracy) with empirical results, ensuring robustness.
   - Explores **data distribution effects** (e.g., power-law sampling, noise magnitude) on circuit formation, deepening understanding of ICL’s dependency on data properties.

3. **Insightful Analysis**:
   - Demonstrates that **multi-head attention** enables parallel circuit specialization, explaining smoother accuracy curves in practical LLMs.
   - Connects findings to prior work on task vectors and redundancy in induction heads, bridging gaps in mechanistic interpretability.

4. **Clarity**:
   - Figures (e.g., attention maps, metric plots) effectively visualize circuit transitions.
   - Appendices provide thorough derivations (e.g., theoretical accuracy) and extended experiments (e.g., multi-head attention).

### **Cons:**
1. **Scalability and Generalization**:
   - Experiments rely on a **simplified model** (2-layer transformer), raising questions about applicability to deeper architectures or LLMs. While connections to LLMs are discussed, empirical validation in larger models is lacking.

2. **Theoretical Depth**:
   - The theoretical analysis in Section 4.3, while validated, is limited to specific conditions (e.g., \(T=2\)). Broader implications for general task inference could be explored.

3. **Comparison to Existing Work**:
   - Limited discussion of how the proposed circuits relate to **other circuit-discovery methods** (e.g., automated circuit finding tools like *Conmy et al., 2023*).

4. **Reproducibility**:
   - Some implementation details (e.g., circuit masking in controlled pruning experiments) are relegated to appendices, which may hinder replication.

---

### Decision · Program_Chairs · 2025-03-04

Accept